# Public engagement for the conduct of a controlled human infection study testing vaccines against *Necator americanus* (hookworm) in areas of active hookworm transmission in Brazil

**Luciene Barra Ribeiro**[1]*, **Andréa Gazzinelli**[1], **Helton da Costa Santiago**[2], **Jacqueline Araújo Fiuza**[3], **Lucas Lobato**[4], **Rodrigo Correa de Oliveira**[3†], **Maria Flávia Gazzinelli Bethony**[1]

1 School of Nursing, From the Federal University of Minas Gerais, Belo Horizonte, MG, Brazil, 2 Department of Biochemistry and Immunology, Institute of Biological Sciences, Federal University of Minas Gerais, Belo Horizonte, MG, Brazil, 3 Group of Cellular and Molecular Immunology- Instituto René Rachou/ Fiocruz Minas, Belo Horizonte, MG, Brazil, 4 Clinical Hospital /UFMG-EBSERH, Belo Horizonte, MG, Brazil

† Deceased.

* lucienebarraribeiro@yahoo.com.br

## Abstract

Controlled Human Infection Models (CHIS) involve administering human pathogens to healthy participants in controlled medical settings, which can elicit complex bioethical issues. Understanding how the community perceives such studies can significantly increase the participant's sense of cooperation and increases the researcher's and the participant's transparency. The current study describes the development of an educational intervention to achieve these ends as it aims to (1) analyze perceptions of the Controlled Human Infection Studies (CHIS), and (2) evaluate the participants' comprehension of the CHIS. **Methods:** This is a qualitative action research that includes the development of an educational intervention with residents of a rural area in Minas Gerais, Brazil, where there is continuous natural transmission of the human pathogen *Necator americanus* ("hookworm"). In this area, it is intended to carry out a proposed phase 3 vaccine clinical trial in the future to test the efficacy of hookworm vaccines using controlled human infection. Two data collection strategies were used: an educational intervention and a focus group. **Results:** The participants' perceptions showed distinct perspectives on CHIS. On one side, they recognized that the investigation is essential for the community, but on the other side, they thought that there would be resistance to its conduct by fear of infection. The idea that the study would generate a benefit for the greater good, contributing to the prevention of hookworm infection, was clearly stated. The participants perceived that the study offered concrete risks that could be reduced by constant monitoring by the researchers. They also mentioned the importance of access to information and the positive influence those who express interest in participating in the study can exert in the community. In relation to comprehension the participants memorized the information, mobilized it to explain everyday situations and created

**Data Availability Statement:** All relevant data are within the manuscript and its Supporting Information files- Data File.

**Funding:** The authors received no specific funding for this work.

**Competing interests:** The authors have declared that no competing interests exist.

strategies to disseminate the study and engage the community in its development. By repeating and making sense of the information, the participant not only assimilates the knowledge transmitted, but also creates new knowledge. **Conclusion**: We concluded that an educational process of discussion and dialogue around participants' perceptions about the CHIS, promotes understanding and allows ways to disseminate information about the research to be collectively created.

# 1. Introduction

Controlled Human Infection Studies (CHIS) involve the administration of a dose of a human pathogen to individuals. These studies are, for a variety of reasons, currently used to (a) expand knowledge about the pathogenesis of infectious diseases, (b) expand understanding of the immune response to such infections, and (most commonly) evaluate the efficacy of drugs and vaccines against such pathogens, as they are more efficient, rapid, and cost-effective manner compared to traditional clinical trials [1–7].

In Europe and the United States of America (USA), numerous CHIS has already been conducted among healthy volunteers for influenza [*Influenza A*], malaria (*Plasmodium falciparum*], cholera [*Bacterium vibrio cholerae*], typhoid fever [*Salmonella Typhi*], dengue [*dengue virus- DENV*], noroviruses [*Norovirus*] and cryptosporidiosis [*Cryptosporidium parvum*], among other pathogens, including hookworm using this method [5,8].

Despite their importance and usefulness, CHIS are a recent development in countries where these pathogens are endemic, such as Brazil, where many vaccine-preventable diseases are found among these low-and middle-income populations (LIMCs). Different biological and environmental conditions, including population genetics, pathogen genetics, and previous infection, can influence the outcome of these studies in ways much different than when conducted in participants from areas where the pathogens are endemic [9,10].

The use of CHIS in Brazil is recent and must conform to established resolutions and guidelines for clinical research development as they are considered a clinical trial and consequently must follow the rule and regulations for such human intervention. However, as they are quite an innovative methodology, they encounter resistant researchers and regulatory agencies. Indeed, CHIS evolve controversial ethical issues considering that healthy participants are infected with pathogens which have deleterious effects. As a result, regulatory processes and specific ethical guidelines need to be discussed collectively, and actions to educate regulatory agents need to be implemented to guide these studies in the Brazilian context [8,11,12].

One of the main concerns for CHIS in LMICS is that many potential participants are considered "vulnerable" by ethical committees, i.e., they lack or have limited education (including illiteracy), poor socioeconomic conditions, and limited access to health care, all of which may influence their decision to participate in CHIS. In Brazil, there is also some evidence of their limited knowledge to distinguish between health research (i.e., CHIS) and standard health care [13,14].

Public engagement is a potent pathway to study understanding of CHIS, developing participant autonomy in making decisions about study participation, and increasing trust and partnership between researchers and the public [12,15–17]. Paying attention to the public, particularly those potentially involved in the research, concerning their perceptions of the study and how it is developed, results in a more inclusive way to produce guidelines that consider the reality of developing countries and help researchers, members of Research Ethics Committees (RECs), and the public understand and engage with CHIS [4,18].

The initiative to know and engage different public and local stakeholders in CHIS and analyze their perceptions on the subject occurs in some countries in Asia and Africa, where this type of study has been developed [5,10,19,20].

To explore public engagement in CHIS, we developed an educational intervention with residents in a hookworm-endemic area, where previous vaccine clinical trials were conducted for this same pathogen. The objectives of the current study were: (1) to analyze local perceptions of the CHIS concept, i.e., their importance, acceptability, the risks and benefits of CHIS, and the consent process, and (2) to evaluate the outcome of this education intervention on participants' understanding of the study. In proposing the present study, the results may assist review by Research Ethics Committees involving population groups of socioeconomic vulnerability.

## 2. Material and methods

### 2.1 Method, location and participants of the study

The study consists of an action research, a qualitative methodology commonly used in the field of education, which is characterized by not dissociating research and intervention. It refers to the development of an educational intervention, the analysis of which lies in the field of exploring interactions and subjectivity.

The study was developed with residents of Americaninhas and 10 small surrounding rural communities. Americaninhas is an endemic area for hookworm, a soil-transmitted helminth infection. The CHIS is intended to accelerate an efficacy study of a vaccine in the future using controlled human infection. Americaninhas is in the rural region of the municipality of Novo Oriente de Minas, located in the northeastern part of the state of Minas Gerais, 570 km from the capital, Belo Horizonte. It has a population of approximately 1,400 inhabitants. In this region, the researchers of the present study have been developing clinical trials for vaccine testing and other investigations for more than 20 years, in a clinic built for this purpose.

The study participants were selected using non-probabilistic sampling. 141 people (approximately 10% of the total population) were invited to take part in the study. This number is justified considering that, in qualitative studies, what matters is not defining a sample with statistical power, because in order to access the subjective aspects of the participants' thinking, it is necessary to use in-depth analysis of the interviews.

The inclusion criteria for the participants were: living in the endemic area (Americaninhas and the surrounding rural area), being 18 years old or over, expressing an interest in this study, being available to take part on the scheduled days and times of the meeting groups and expressing an understanding of the Informed Consent Form (ICF). Residents who worked during the scheduled times or had previously scheduled appointments were excluded from the study. This study did not include minors.

Participants were invited to take part in the intervention through home visits by members of the research team. Those who were willing to participate went to the clinic in Americaninhas, where all previous clinical trials conducted by the project team took place. Transportation to the clinic was provided by the project coordinators.

### 2.2 Data collection, recording, and analysis

As data collection strategies, an educational intervention and a focus group were used. The intervention was carried out on May 26, 27 and 28, 2022, through nine meetings lasting one hour each and including 6 to 13 participants (residents and community representatives). Subsequently, a focus group was held with ten participants drawn from the members of the educational intervention groups. The meetings that comprised the education intervention and the focus group were videotaped with prior authorization from the participants.

The educational intervention, composed of multiple resources, had the double role of educating and favoring access to more spontaneous elements of the participant's behavior and perception. The focus group, developed after the end of the intervention, whose main feature is to encourage an informal conversation among the participants, without the researcher's constant interference, was intended to spark discussion around the issue of: *"What do you remember about the explanation of the CHIS?"*

Data analysis was analyzed using Bardin's Content Analysis [21]. Participants' questions, answers and comments were used as the corpus of analysis. Content analysis was carried out on all the perceptions produced by the groups as a whole, to the detriment of individual perceptions. In this way, consistency with the objectives of the study and the chosen research methodology would be preserved.

In order to analyze perceptions of the study (objective 1), thematic axes or categories were created beforehand, prior to processing the data. As they are closely linked to the content around which we sought to collect the participants' perceptions, these categories guided the conduct of the study, as well as the intervention and focus group scripts. Five thematic axes were defined: acceptability, study characteristics, importance, risks and benefits and community information.

After transcribing the footage of the intervention and the focus group, the material was read with the aim of intensive immersion in the data and coding by the researchers. The coded speeches of the participants were then grouped into thematic axes/categories. To ensure the accuracy of the coding, the content analysis process was characterized by periodically returning to the data and progressively refining the categories. This coding was carried out by three independent researchers.

The next step was to break down the data from each thematic axis/category into record units, the smallest units of meaning in the participants' speeches. To identify them, they were grouped according to the similarities and common aspects between the ideas contained in the speeches. The nature of the research problem and the objectives of the study were the aspects used to define each of the recording units.

Table 1 shows how this process of extracting units of meaning from the speeches was carried out. The category that served as an example, for illustrative purposes only, was acceptability. The lines in the table represent the groups of participants (Group 1, group 2 and group 3). In each group, the participants' speeches regarding acceptability were marked with color according to the identified registration units.

The context units were subsequently defined, superior to the registration units, which allow understanding the broader meaning of the items obtained, placing them in their context.

The same process was used to analyze the understanding of the study (objective 2). However, the categories emerged from the data itself; they were not established a priori and were unpredictable. These categories consisted of the cognitive processes produced by the participants during the intervention and focus group interactions. An expanded notion of cognition was used, which goes beyond the process of solving problems, but is defined as the invention of oneself and the world [22].

## 2.3 The educational intervention

The problematizing and aesthetic approaches were used in the educational intervention. Problematization refers to dialogue with others and with reality through the production of meaning. In the aesthetic approach, the emotional and practical dimensions of the participants' experience are valued, as well as subjective production [23]. The following educational strategies were carried out: (1) conversation circle (2) the explanatory video (3) practical demonstration e (4) storytelling.

**Table 1. Participants' perceptions (discourse) on the acceptability of EIHC.**

| Group | Participants' perceptions (speech) | | |
|---|---|---|---|
| G1 | Pa: I think the community will be a little afraid at first because. . . they're going to say "Oh, but it will put the disease in us", I can hear people saying that it will put the disease in us. But, for example, what am I going to say when someone from the community comes to talk to me about Fiocruz wanting me to take part in a vaccine or research project? | Pa: I think that when I talk to people, like she said, some will have doubts and others will be afraid, right? And there's the question of those who will understand and think it's good for the place, especially, as was said, in our region, which is very in need of sanitary treatment. So, looking at the good side of the research, many people will like it. | Pa: I think this issue of developing a vaccine for it is very valid. But firstly, something that goes beyond that: I think basic sanitation is a social problem of ours. This is very far from the reality for many families. |
| G2 | Pa: Oh, I think they'll be fine with it. Because the region needs it. There are many cases. | Pa: "Someone from my community spoke to me. What is this vaccine? Then I said, "There are many good things to come from this; I've always participated directly; I do, don't I? Then she said, "Ah, so that's good. So, if you know that you're listening, tell me. That's fine. It's very good there. I've enjoyed it ever since I joined, haven't I? I really enjoy taking part too, I'm always very well looked after, thank God, on time." | Pa: "I think people will accept it, considering the difficulties of treatment in the region." Pa: Because the staff are very well-trained and informed! It's. . . there's a very good structure, isn't there? In terms of doctors, nurses, and a laboratory, there's everything. The people, especially the people from here in Americaninhas and the region, are. . . there are many people who would like to take part, aren't there? And sometimes they don't have the opportunity to take part. I think that's going to be easy for us here. God willing. |
| G3 | Pa: "But that's very good; we participate, don't we? Because if we participate and vaccinate, we know we're helping to create a vaccine, so it's a very important thing, I think. I don't talk much; I'm too shy to talk. But I think it's very important. I've been here many times, and I've had a few vaccinations there. But the team takes care of everything properly; they pass it on to the people, don't they? That's when many people downplay it, say it's nonsense, but when we explain it to them, many even are keen to join the project too." | Pa: About fear, there are people who even hide so as not to be invited. I've even heard it, since we're taking part here: I already ran from those people when I saw the car. They tell me that they're going to take blood or that they will have to take blood. They say, I don't know what they're taking that blood for. But this fear arises, I think, because of something we've heard. Since there are such things. I think that's why. | Pa: I really like taking part; I've already taken part here once, and I really like the fact that, as she said, sometimes we won't see the results, but other lives will receive the results. I really enjoyed taking part as a volunteer in the project, and if God helps it work out, I'd like to take part again. |

Legend of recording units: The color red represents the *fear of infection*, the blue color represents *trust in the team*, the green color represents *vaccine production* and the yellow represents *good for the community*. And the abbreviation Pa is *participant*.

The conversation circle, contextualizing hookworm disease, and the CHIS planned to be developed in the community prioritized orality. Understood as verbal expression, orality is an experience that activates its own meanings and aspects of people's cultural universe. Researchers and participants exchanged knowledge about the illness without hierarchizing. The participants told their experiences of living with the disease for some time. When one chooses to talk "with" the other instead of "to" the other, interaction is achieved, knowledge is open to confrontation, and bonds are built between researcher and participant.

The explanatory video explained how the CHIS with hookworm is developed and its importance. Through drawings with realistic strokes and vibrant colors, those involved could look at the place where they live in a less repetitive and naturalized way and, with that, problematize the environmental and social conditions of the region.

The demonstration was how to illustrate the controlled infection procedure and how the clinical trial would be conducted, including the groups that would receive the vaccine and the placebo. The participants were invited to volunteer to simulate the larvae inoculation.

By mixing reality, fiction, and playfulness, the story focuses on the daily life of the residents of Americaninhas and the strangeness of the community with the unexpected arrival of the researcher. Through storytelling, without additional resources, using only voice and body, the researcher invited the participant to enter the narrative, project himself into the characters, and express himself freely.

### 2.4. Ethical aspects

Participants were informed about the study verbally and had the opportunity to read the printed ICF, which was given to each guest. After reading it and resolving any doubts, those who decided to take part signed and initialed all the pages of the physical document in two copies of equal content. The researcher also signed both copies of the ICF. One copy was given to the participant and the other was filed with the project documents at the Clinic. The entire consent process was described on an addendum sheet and attached to the ICFs and filed in the study folder.

The study was conducted based on resolution 466 of 2012 of the National Health Council of Brazil, which approves the guidelines and regulatory standards for research involving human subjects. It was submitted to the Ethics and Research Committee of the Federal University of Minas Gerais and approved under CAAE: 57040322.4.0000.5149 and consubstantial opinion number: 5.412.848 on May 17, 2022.

## 3. Results

A total of 95 residents participated in the study, the majority being women (n = 62), with a mean age of 37 years (18–55 years). The participants' occupation was mainly housekeeper (n = 30) and farmer (n = 27). Other occupations were cowboy, beekeeper, municipal civil servant, teacher, student, and service provider. Of the 95 participants included, 28 had already participated in clinical research.

The results were presented in two parts: the first includes the perceptions of the study participants, categorized into thematic axes previously defined. The results regarding the participants' comprehension of the CHIS were described in the second part of the study.

### 3.1 Participants' perceptions of CHIS

The participants' perceptions of CHIS were categorized into five thematic axes, as represented in the Fig 1.

**3.1.1. Acceptability or desire to participate.** When asked about the community's perception of the study, most participants expressed that there would undoubtedly be distinct positions. On one side, there was the recognition that the investigation would be necessary for the community, and on the other, there would be resistance to its development.

> *"Many people will criticize, but many will think it is good because it is a way to help the community."* (G7)

> *"People who think it will be good will participate, but others will have a little difficulty and may deny it."* (G6).

As a reason for resistance to participation, residents mentioned uncertainty about the study, fear, and concern about becoming infected with the worm, especially if *"considering the difficulties of treatment in the region."* (G4). Even admitting these feelings, some participants bet on the approval of the study.

> *"I think some people will have doubts, others will be afraid. And there are those who will understand and think it is good for the place."* (G8).

They also reported that *"there are people who even hide not to be invited because of what they hear."* (G4). In this case, the allusion is to the knowledge commonly present in the collective imagination in clinical research related to blood collection and its motivations.

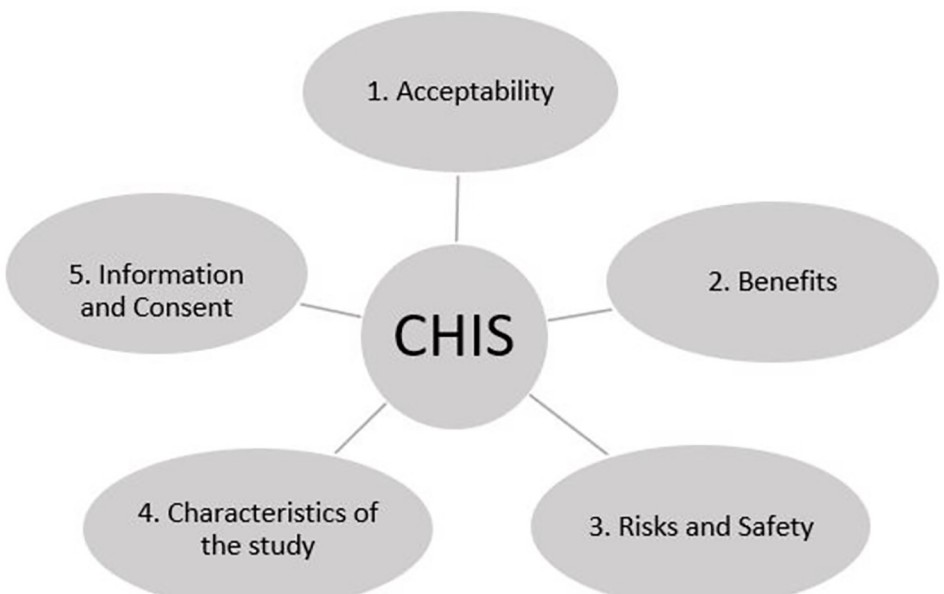

**Fig 1. Thematic axes- the thematic axes represent the 5 categories of participants' perceptions in relation to CHIS.**

**3.1.2. Benefits.**   The idea of benefit for the greater good was shared by several participants, who considered participation as the critical altruistic motivation to contribute to disease prevention and cure. Emphasis was given to the benefit of the study for the collectivity.

*"In a way, by participating in the research, we are helping other people and the community. The vaccine will not only be good for me. It will be useful for so-and-so, so-and-so."* (G3).

"It is not simply being a guinea pig, but from the moment we were tested, we were volunteers, it is the same as if we were donating ourselves. It would be for the good of many there. For example, this group here, about 10, passing the test, and the vaccine being approved; imagine how many will benefit? How much good will be done through these people?

There is a perception that the benefit of the study is short-term and that in the region where they live, advances are slow and take years to happen. A vaccine like this could be developed more quickly and would, if successful, benefit the future generation.

*"In our region, a change takes at least 100 years. I have clarified it for my family members, and they understand that it will be good for us, but especially for future generations. The vaccine that will be developed may not be for me, but for my son or my grandson."* (G9).

**3.1.3. Risks and security.**   In the perception of the participants, if, on the one hand, the study offers substantial risks to the volunteer, on the other, this risk, when monitored, is effectively reduced. Therefore, it can be affirmed that there is a rationalization of troubles.

*"They give the treatment before reaching the risk because we will be monitored, and before reaching the anemia phase, they treat and give albendazole."* (G4).

*"There is a risk. But you will be supported, you will have the treatment, you will have the follow-up. There is nothing that will be good that doesn't have risk. In everything we do there are risks".* (G4).

Risk, for the participant, is inherent to human existence. The indefinite pronouns all and nothing are used to naturalize the risks considered acceptable because of the presence of treatment. Follow-up care and treatment, in turn, play an essential role in minimizing risk.

The natural risk of contracting the disease is also present among the participants who highlighted the chance of being naturally infected by hookworm and developing the symptoms. However, they affirm the impossibility of death. Thus, the risk is not absolutized.

*"But it's no different from when you catch it anywhere and get infected because you're going to be treated the same way."* (G4).

The study is considered safe and reliable by the participants because it offers no risk of death and the presence of multi-professional assistance (doctors, nurses, and researchers).

*"It is a worm being researched, and they are trying to develop a vaccine. This disease does not cause death, and the person will be followed and monitored by professionals. That gives more confidence."* (G6)

*"They will do all kinds of tests, and if the person is contaminated, they do the treatment."* (G2)

*"You're going to have the worm, but you will have the medicine to kill it, and even the person infected with the worm will have the follow-up."* (G1)

Some of the participants, when they trust the researchers and their power analogous to that of doctors, end up underestimating the risks of study participation:

*"Let's assume. . . if the person decides to participate, he will be accompanied and will not be at risk of dying. Because if there were a risk of death, nobody would put anybody's life at risk. So, it's a safe thing. And the disease has treatment."* (G6).

Using the indefinite pronoun no one twice in the discourse gives the study an invariable character of safety. However, the guarantee of complete security in clinical research is nonexistent in the realm of the imaginary and the possibility of the absolute.

**3.1.4. Survey characteristics.** For participants, in general, the CHIS is perceived as a vaccine test in which individuals are infected:

"In the test, people will be infected purposely, there will be the vaccination and, if the vaccine does not work, they will get the infection" (G8).

*The objective of the test for the participants was to evaluate the effectiveness of the vaccine:* "it is to know if, in the person who took the vaccine, the larva will get in" *and* "identify if the vaccine works" *(G7). Thus, participants understood that a challenging experiment tests the efficacy of a vaccine to prevent infection and disease in a small group of volunteers.*

*"They will wash the larva well and put it on your skin, and then do a test to see if the vaccine works. What can happen is that if the vaccine doesn't work, the person will get infected and will have to be treated".* (G2)

The perception is, therefore, that the study refers to a vaccine test and, for this reason, has no therapeutic purposes. The participants highlight the purposeful character of the infection that occurs through the inoculation of the larva. They recognize that not everyone will receive the vaccine, a condition that refers to the concepts of placebo and randomization.

*"We talked about the vaccine test, how the inoculation will be done: it puts the patch on the skin purposely, but first some people will receive the vaccine and others will not"* (G2).

*"To do the tests, make the vaccine application. Some will get the real vaccine, others won't. Then you put the larva in, and it will stay for an hour in observation."* (G8)

An essential feature of the vaccine has also been identified, which is to accelerate the process of developing a vaccine.

*". . .if the vaccine comes fast, it will be good [. . .] because, in fact, it is like you said: developing a vaccine is not easy. It takes a long time, but if there is a technology that can come out faster, that would be nice*! (G7)

*[. . .] it will be research that can accelerate this, the vaccine will be approved faster and will not need many people [included in the study]"* (G6).

**3.1.5. Information and consent.** As ways to face fear and uncertainty, participants mentioned the importance of access to information and the positive influence that those who express interest in learning about and participating in the study can exert on the community. Thus, circulating that individual interest in learning about the study is essential for the construction, in the collective, of a positive view about it:

*". . .we go and explain what it is, and that there will be an accompaniment. Surely, afterward, the person has that understanding too"* (G2).

*". . .knowing is fundamental; you must have adequate knowledge about the study."* (G6).

On the other hand, the lack of knowledge about the study, as well as the resistance to the new would be responsible for the difficulty in approaching the study:

*"People might think it might not be good for the region since they have many doubts about a different thing coming into the place. . .. And it takes a while to make people understand."* (G7).

Just as the optimistic view of some encourages participation, there are those who, in the perception of the participants, discourage others from participating. The shock and strangeness of the new and different may prevent some residents from readily welcoming the study.

*"In our region here, there are people who are ignorant and take others out of their heads even to come in the meeting like this one here."* (G6).

In these circumstances, it is best to avoid being carried away by common opinion, as one participant stated, highlighting the importance of participation and involvement in the study to construct a critical consciousness about it.

**Table 2. Chart of results including categories, recording units and context.**

| Categories | Recording unit | Context unit |
|---|---|---|
| Acceptability | Good for the community | The study is acceptable because it is important for the production of a vaccine and is good for the community. There is trust in the team of researchers because of the seriousness of their work, as seen in previous studies. At the same time, the community may resist the study, due to fear of infection and the vaccine. |
| | Fear of infection and vaccine | |
| | Trust in the team | |
| | Vaccine production | |
| Benefit | Helping the community in prevention | The study helps the community to prevent the disease. The benefits are not immediate and contribute to future generations. As for the risks, they are reduced due to the presence of a multi-professional team and follow-up in the study. |
| | Contribution to the future generation | |
| | Following up the study | |
| Risk and e Security | Monitored risk and treatment | There are risks of infection in the study and therefore fear of getting sick if the vaccine doesn't work. The side effects of the vaccine are also risks offered by the study. However, there is a perception that the study is safe, considering that the disease is treatable, does not cause death and there is constant monitoring. |
| | Does not cause death | |
| | Risk of infection | |
| | Risk of vaccine side effects | |
| Study characteristics | Vaccine test | The study consists of testing vaccine efficacy, which can generate immunization, thus protecting the community. It is characterized by controlled infection and rapid vaccine testing. |
| | Controlled infection | |
| | Rapid vaccine testing | |
| | Analysis of vaccine efficacy | |
| | Immunization | |
| Information and consent | Explanation/clarification | Transmitting the explanations to the public and adapting the language to the public are ways of informing the community and promoting understanding of the study. There is a need for active participation in meetings with the researchers in order to access the explanations of the study and the views of other participants. The community has both a positive and negative influence on the participants' perception of the study. |
| | Participation in meetings | |
| | Community influence | |
| | Promotion of understanding | |
| | Suitability of language | |

*"They will say*: *if I were you, I wouldn't participate. What are you going to do there*? *What about the contamination of this worm*? *You can get infected and have a bigger problem*! *Because we always hear words to take us out of the focus. So, we must participate [in the meetings] to see how it is.*" (G6).

Each thematic axis presented was organized based on the recording units which referred to the meanings expressed by the participants during the interactions that took place in the meetings. When shared in the group, these meanings are reinforced, denied or affirmed. When grouped and placed back into the discourse, the registration units can also be understood as a unit of context. Table 2 shows the categories and recording units that make up the study results of the study.

### 3.2 The participants' understanding

The engagement intervention favored the participants' understanding of the CHIS. In the interactions during the intervention, the following cognitive processes were expressed by the participants: memorization skills, problems with reality, production of meaning for the information, and an inquiring attitude.

After experiencing the education, the participants demonstrated that they had memorized information, mobilized this information to explain everyday situations and were able to create strategies to disseminate the study and engage the community in its development. By repeating and making sense of the information, the participant not only assimilates the knowledge transmitted, but also creates new knowledge. The memorization of the information was observed

when the participants, prompted, were able to repeat exactly the researchers' explanation of how the infection with the larva occurs.

Surprising questions that, as a rule, are only asked by those who effectively understand the content emerged among the participants. As an example, we can cite the questioning about the imminent risk that, with the study, the infection *"ends up happening to everyone"*: *"My question is*: *I come here and contaminate myself with the larva. If I come home and defecate in the field, can someone else get infected*?*"* (G9) Within the same perspective, the question *"if the vaccine has already been tested on animals"* (G3), asked in the intervention, expresses the participant's desire to make sure that the vaccine has the potential to be safe. Among the participants, the question related to whether to participate in the study was also raised.

The problematizing of reality was evidenced when the participants made an approximation of the story told by the researcher with what they experienced in their daily lives. They spontaneously identified with the characters in the story and simultaneously projected themselves into the plot's conflicts because of their proximity to concrete reality. They recounted daily events and, consequently, the problems in which they get entangled in the region: *"I leave home early to kill the ox, and I go with my slippers on. My wife tells me to wear boots, but I feel more comfortable in slippers. Then, when I'm running after the ox, I lasso him, kill him, and keep my feet on the ground. Sometimes I have this disease, and I don't know. I must do the test"* (G9).

From the speech, it can be observed how the issues related to health are not only associated with the adoption or not of appropriate behaviors. This is because the production of health is constantly crossed by routines, interactions, and, above all, the meanings of life. For this participant, *"having one's feet on the ground"* metaphorically has importance for him, as it makes him feel free and *"at ease."*

In talking about the study, the participants also invested in other clippings of the problem. Rather than fixating on pre-existing knowledge, they mobilized the assimilated information from the researcher to create an idea about "how" and "in what way" to engage the community in the study. Recognizing the importance of this process, they considered it essential to address elements that would contribute more directly to people's acceptance of the research. These elements would be the answers to questions related to "why" to do the study and participate in it. In this way, the information that justifies the study would be privileged, in other words, the notion that this research model would favor more rapidly obtaining results.

To defend the same point of view—the explanation favors the understanding of the study— the participant throws light on the controversy about being a guinea pig or not when participating in a CHIS:

> *"It's not simply being a guinea pig, but from the moment we were tested, we were volunteered; it's the same thing as if we were donating ourselves. It would be for the good of many there. For example, if ten pass the test and the vaccine is approved in this group here, imagine how many will benefit? How much good will be done through these people? So, it is not simply being a guinea pig. It may impact when we speak, but the person will say, "Wow, but injecting?" But, at that moment, we go to these people and explain what it is and that there will be follow-up. For sure, then the person has that understanding too."* (FG)

The participant emphasizes the human contribution or scientific solidarity of the volunteer —fundamental to the development of research and science.

## 4. Discussion

A current concern among researchers, ethics committees, and regulatory agencies about CHIS is the conduct of these studies in vulnerable populations with inadequate access to health

services and limited autonomy and capacity for informed consent. Therefore, it is necessary to ensure the public understands all the information and procedures surrounding these studies for participation in decision-making [7].

The intention of conducting an intervention with different strategies was to identify residents' perceptions of an endemic area about CHIS and to assess whether understanding information about the disease, characteristics, benefits, risks, and the importance of developing this type of study in the locality occurred. Although CHIS are new to the community studied, the perceptions identified showed that the population could accept this type of study satisfactorily. Researchers point out that the residents' perceptions of endemic communities about these studies help the researcher better understand the concerns and how individuals in these localities think about obtaining volunteer participation [19,24].

In this study, participants felt that CHIS using hookworm might be safe because this pathogen will be administered under well-controlled conditions by the researchers. Many considered altruism and a benefit for future generations as a purpose for participating in the study to test vaccines that may prevent future infections. They realize that there are risks of developing symptoms, but they recognize that hookworm is not a disease that induces mortality but mostly morbidity and that there is a "rescue drug" (albendazole) that can be administered which effectively removes the infection, with few side effects. On the other hand, participants were concerned about reactions from people in the community about "*purposely infecting a healthy person.*" Several studies indicate that this is not uncommon, and not only in populations from low- and middle-income countries [9,19,13,25]. Even in studies involving diseases that carry higher risks to participants, such as SARSCoV2, despite the concerns, the general manifestation is that the study should be developed [20].

Adequate information was perceived as a condition to favor acceptance by the community. In this regard, participants affirmed the importance of attending meetings with researchers to avoid (mis)information and defined contents to be emphasized to establish community trust patterns. Unlike the usual perception in the research field that it is the researcher's job to engage the public, the participant cogitated their responsibility to take the job of informing the community into their own hands. It is considered today that it is not only the researcher who needs to be involved in informing and engaging the public and society. The participants must be affected and made aware to become active participants in this process because the engagement refers to a process that is based on the concern to ensure, more democratically, the active participation of the different public in decision-making related to involvement in research and its implications on the collectivity [4,26–28].

The perceptions of the EIHC, when expressed and shared in the interactions that took place during the intervention, favored the understanding of the study. In contrast to modern cognitivist science's notion of understanding, the study adopts the concept of understanding as a simultaneous movement of recognition (repetition e reproduction of information) and invention (creation). The cognitive processes expressed by the participants affirm this double movement [23].

Community engagement in research has recently been seen as essential to the development of clinical trials and to the consent process, especially in the case of CHIS. Studies indicate that it influences how individuals understand risks and benefits and fully understand informed consent for free decision-making [18,24]. Moreover, they ensure transparency of the whole process and favor building a trusting relationship between the parties [19].

## 5. Conclusion

In this study, educational intervention strategies favored understanding the study. By including dialogue and valuing the singular and collective experiences of the individuals, the

intervention allowed the participants to produce meaning for the information transmitted by the researcher.

The participants' perceptions indicated that the participant, rather than simply assimilating, understood the information. Creating a problem from a theme, not just solving it, is a necessary condition to define if something has been understood. By seeking to probe how reliable the study and the vaccine are, the participant denoted openness to consider participating.

The involvement and engagement of the participants, beyond understanding, were other significant results obtained from the education intervention. Participants felt involved in the work of talking to the community about the study. They recognized their important role in exchanging experiences and information with the community in a movement of sharing knowledge and power with the researcher.

It is concluded that the participatory and educational process of approaching community perceptions promotes engagement and allows the creation of the best ways to disseminate information about the research and formulate educational materials that consider the local context.

## Supporting information

**S1 File. Each Excel worksheet represents: 1- the number of participants involved in this research.** Number of participants included and not included in this study, the gender of the participants and their occupations; 2-Type of intervention carried out in the education action; 3- The categorization of the participants' statements according to the thematic axis of the study; 4-The answers to the question that guided the focus group: "What do you remember about the explanation about the CHIM?".
(XLSX)

## Acknowledgments

Dr. Rodrigo Correa de Oliveira passed away before the submission of the final version of this manuscript. Dr. Luciene Barra Ribeiro accepts responsibility for the integrity and validity of the data collected and analyzed.

## Author Contributions

**Conceptualization:** Andréa Gazzinelli, Rodrigo Correa de Oliveira, Maria Flávia Gazzinelli Bethony.

**Data curation:** Luciene Barra Ribeiro, Andréa Gazzinelli, Helton da Costa Santiago, Jacqueline Araújo Fiuza, Maria Flávia Gazzinelli Bethony.

**Formal analysis:** Luciene Barra Ribeiro, Andréa Gazzinelli, Helton da Costa Santiago, Jacqueline Araújo Fiuza, Lucas Lobato, Rodrigo Correa de Oliveira, Maria Flávia Gazzinelli Bethony.

**Funding acquisition:** Rodrigo Correa de Oliveira.

**Investigation:** Maria Flávia Gazzinelli Bethony.

**Methodology:** Luciene Barra Ribeiro, Andréa Gazzinelli, Helton da Costa Santiago, Jacqueline Araújo Fiuza, Maria Flávia Gazzinelli Bethony.

**Project administration:** Rodrigo Correa de Oliveira.

**Supervision:** Maria Flávia Gazzinelli Bethony.

**Writing – original draft:** Luciene Barra Ribeiro, Andréa Gazzinelli, Helton da Costa Santiago, Jacqueline Araújo Fiuza, Maria Flávia Gazzinelli Bethony.

**Writing – review & editing:** Luciene Barra Ribeiro, Andréa Gazzinelli, Helton da Costa Santiago, Jacqueline Araújo Fiuza, Lucas Lobato, Rodrigo Correa de Oliveira, Maria Flávia Gazzinelli Bethony.

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
