## [Decision Letter · Decision Letter 0]

19 Oct 2023

PONE-D-23-25173Public Engagement for the Conduct of a Controlled Human Infection Study testing vaccines against Necator americanus (hookworm) in areas of active hookworm transmission in Brazil.PLOS ONE

Dear Dr. Ribeiro,

Thank you for submitting your manuscript to PLOS ONE. After careful consideration, we feel that it has merit but does not fully meet PLOS ONE’s publication criteria as it currently stands. Therefore, we invite you to submit a revised version of the manuscript that addresses the points raised during the review process.

We look forward to receiving your revised manuscript.

Kind regards,

Kenneth Bentum Otabil, PhD

Academic Editor

PLOS ONE

4. Please amend the manuscript submission data (via Edit Submission) to include author Rodrigo Correa de Oliveira.

5. We note that Figures 1 and 2 in your submission contain [map/satellite] images which may be copyrighted. All PLOS content is published under the Creative Commons Attribution License (CC BY 4.0), which means that the manuscript, images, and Supporting Information files will be freely available online, and any third party is permitted to access, download, copy, distribute, and use these materials in any way, even commercially, with proper attribution. For these reasons, we cannot publish previously copyrighted maps or satellite images created using proprietary data, such as Google software (Google Maps, Street View, and Earth). For more information, see our copyright guidelines: http://journals.plos.org/plosone/s/licenses-and-copyright.

1. You may seek permission from the original copyright holder of Figures 1 and 2 to publish the content specifically under the CC BY 4.0 license. 

Reviewers' comments:

Reviewer's Responses to Questions

**Comments to the Author**

1. Is the manuscript technically sound, and do the data support the conclusions?

Reviewer #1: Partly

Reviewer #2: Partly

Reviewer #3: Yes

2. Has the statistical analysis been performed appropriately and rigorously? 

Reviewer #1: No

Reviewer #2: No

Reviewer #3: No

3. Have the authors made all data underlying the findings in their manuscript fully available?

Reviewer #1: Yes

Reviewer #2: Yes

Reviewer #3: Yes

4. Is the manuscript presented in an intelligible fashion and written in standard English?

Reviewer #1: Yes

Reviewer #2: Yes

Reviewer #3: Yes

5. Review Comments to the Author

Reviewer #1: ABSTRACT

Aim:

Under the abstract, the first (1st) aim has to be revise to enhance clarity and understanding of the aim. The sentence is too long and wordy. (Lines: 20, 21, 22)

Methods:

The first line is too long and wordy. Please this should be broken into two sentences to enhance clarity and comprehension. (Lines: 23, 24, 25, 26, 27)

Results:

i] Please revise the sentence: “Concerning the risks of CHIS ….. is reduced.” It is not clear. (Lines: 32, 33, 34)

ii] Please revise the paragraph on engagement intervention (Lines 36, 37, 38). The abstract needs a summary of the results of the engagement intervention. The sentence provided only information on how the researchers assessed participants’ understanding of the CHIS but not the results of the engagement intervention.

INTRODUCTION

The abbreviation CHOMs has to be written in full in its first mention. (Line 64)

The Educational Intervention

The first sentence (paragraph) has to be revised for clarity and comprehension. (Lines: 132, 133, 134)

“Orality” and “Problematization” have to be defined in context to this research for comprehension by the reader. The reader has to understand what the researchers meant by “Orality” and “Problematization”. Please revise. (Lines: 138, 139)

RESULTS

For the various issues addressed in this section, numerical presentation of the results has to be given.

E.g. Considering “Acceptability or Desire to participate”, the results have to present to the reader how many participants of the 95 expressed distinct positions?

And this goes for all other sections under results.

Please also worth of note, are the definitions for G1, G2, G3, G4 – G9. These have to be given to enhance comprehension of data.

Thanks.

Reviewer #2: The manuscript was well written and scientifically sound, However the authors had these few issues to address

1. Provided result on the demographic and clinical characteristic of the study participants in the manuscript

2. Under methodology, describe the method use to identify the hookworm infection

3. Provide information on the ethical approval for the study

4. Provide information on the inclusion and exclusion criteria used for the study.

5. Provide a flow chart for the study showing the number participant recruited to the end of the study. provide information on those who exited from the follow and the reasons.

Reviewer #3: The Editor

Committee for Human Research and Ethics

University of Energy and Natural Resources

Dear Sir/Madam,

RE: INVITATION TO ACT AS A PEER REVIEWER

This report is submitted in response to your request to peer review the underlisted manuscript’s suitability for publication in PLoS ONE journal.

Manuscript Number: PONE-D-23-25173

Manuscript Title: Public Engagement for the Conduct of a Controlled Human Infection Study testing vaccines against Necator americanus (hookworm) in areas of active hookworm transmission in Brazil.

Authors: Luciene Barra Ribeiro. Andréa Gazzinelli. Helton da Costa Santiago. Jacqueline Araújo Fiuza. Lucas Lobato. Rodrigo Correa de Oliveira. Maria Flávia Gazzinelli Bethony.

Article type: Research Article

The manuscript satisfies all the seven (7) criteria for publication in the PLOS ONE journal, however, the following comments and suggestions below are for the authors' consideration.

The statistical analysis performed was not appropriate and rigorous because even though the researchers used Bardin's Content Analysis for their data analysis but failed to categorize or codify the words, themes, and concepts within the text that would enable them to subject their data to quantitative analysis. Content analysis can be both quantitative (focused on counting and measuring) and qualitative (focused on interpreting and understanding). In both types, you categorize or “code” words, themes, and concepts within the texts and then analyze the results. It is therefore recommended that the authors do the needful and subject their data to quantitative analysis.

Line 4: make 1 attached to author Andréa Gazzinelli superscript

Line 24: Delete the definite article “the” that precedes ongoing natural transmission

Line 26: Add “is” to before the first word “targeted”

Line 49: Change the word “has” that precedes already to “have”

Line 53: Delete phrase “have been conducted” for repetition in line 49 and 50

Line 59: Change participation to participants and delete “in” that precedes areas

Line 64: Replace CHOMS with CHIS

Line 65: Replace the word “now” with “which”

Line 78: Replace “resulting” with “results”

Line 108: Replace the word “among” with “from”

Line 125: Define TCLE before you abbreviate it

Line 126: Define the age limits of minors in Americaninhas and re-caste the last sentence (e.g. This study did not include minors).

Line 127: Which body promulgated resolution 466 of 2012?

Line 175: Separate goodwill into “good” and “will”

Line 216: Highlighted

Line 346: Re-caste (i.e. residents' perceptions of CHIS in an endemic area)

Line 354: Delete “thought”

Line 358: Replace “with” with “which”

Re-draw figs. 1 and 2 to show at least the North Pole, and scale.

Thank you.

Dr. Francis B. D. Veriegh

6. PLOS authors have the option to publish the peer review history of their article (what does this mean?). If published, this will include your full peer review and any attached files.

Reviewer #1: **Yes: **SETH TENKORANG BOATENG

Reviewer #2: No

Reviewer #3: No

---

## [Author Response · Author response to Decision Letter 0]

17 Dec 2023

PONE-D-23-25173

Public Engagement for the Conduct of a Controlled Human Infection Study testing vaccines against Necator americanus (hookworm) in areas of active hookworm transmission in Brazil.

Dear Editor and Reviewers,

Please find enclosed a revised version of the manuscript that addresses the points raised during the review of PONE-D-23-25173 “Public Engagement for the Conduct of a Controlled Human Infection Study testing vaccines against Necator americanus (hookworm) in areas of active hookworm transmission in Brazil” by the authors Luciene Barra Ribeiro1. Andréa Gazzinelli1. Helton da Costa Santiago2. Jacqueline Araújo Fiuza3. Lucas Lobato4. Rodrigo Correa de Oliveira3. Maria Flávia Gazzinelli Bethony1.

We have extensively revised the manuscript according to the reviewer’s suggestions. We appreciate the comments as they contributed greatly to improving the quality of the manuscript. Neither the original nor the revised version of the manuscript are under consideration for publication elsewhere. As the corresponding author, I certify that the listed authors participated meaningfully in the study and that they have approved this resubmission.

The following are included:

1-A letter that responds to each point raised by the academic editor and reviewer(s). 

2-A Track Changes version of the manuscript that highlights changes made to the original. 

3-A “clean” (revisions accepted) version of the manuscript. 

Responses to reviewers were prepared based on the article with track changes. To check the correct lines, please check the track changes version

Below are our responses to the editors and reviewers’ comments.

General Comments

The manuscript was revised according to the suggestions of the editor and the three reviewers. 

Editor’s comments: 

The financial disclosure has been changed. Funding from Wellcome Trust Funding was included.

For this project, the deposit of laboratory protocols in protocols.io to enhance the reproducibility of your results is not applicable. Therefore, this recommendation was not applicable.

Journal Requirements

This manuscript follows the submission requirements of the journal. Following the guidelines related to information in the Informed Consent Form (ICF), the consent process and the inclusion and exclusion criteria were added to the manuscript to clarify issues raised in the journal's review of this manuscript. Item 2.4 (ethical aspects) can be found in the methods section. According to the guidelines, the ORCID ID was linked to the manuscript, and the author, Rodrigo Correa Oliveira, was included. Regarding the figures which contain maps and as such require a copyright, the authors have chosen to remove them from the manuscript as they are only illustrative, and their removal will not affect the readers' understanding.

Supporting information is now included.

Comments to the author

The responses to the issues raised in comments numbers one and two are below:

The reviewers requested additional details about the statistical analyses employed. As this is a qualitative study, data were analyzed using Bardin Content Analysis. A paragraph has now been included in the Method Section to clarify the study design and the qualitative analysis methods. Our qualitative study does not seek to quantify or measure events, so statistical methods are not used. Qualitative methods like focus groups are commonly employed in educational studies due to their exploratory nature, which cannot be analyzed statistically.

However, it is essential to highlight that the reviewers' observation regarding the need to define the technical “registration units” significantly improved the text's comprehensibility and increased the transparency taken to reach the study results. Recognizing that they needed to be adequately explained in the manuscript, we rewrote section 2.2, Data Collection, Recording, and Analysis, and part of Materials and Methods. We also added a new figure to facilitate comprehension of the analysis.

Response to Reviewer # 1

1-Abstract: 

1.1-Aim: Under the abstract, the first (1st) aim has to be revise to enhance clarity and understanding of the aim. The sentence is too long and wordy. (Lines: 20, 21, 22).

Done: The objectives have been reworded to make the summary understandable and clear to the reader. The following objectives were included in the article: (1) analyze perceptions of the Controlled Human Infection Studies (CHIS) and (2) evaluate the participants' comprehension of the ICF. (Lines: 21,22,23,24).

1.2-Methods: The first line is too long and wordy. Please this should be broken into two sentences to enhance clarity and comprehension. (Lines: 23, 24, 25, 26, 27).

Done: The first line has been separated in two to understand the text better. Additionally, we have included the information that a Controlled Human Infection study is planned. The objective of the current study was to analyze participants' perceptions and comprehension following an educational intervention. (Lines:24,25,26,27,28,29).

1.3-Results: i) Please revise the sentence: “Concerning the risks of CHIS ….. is reduced.” It is not clear. (Lines: 32, 33, 34)

Done: The sentence has been rewritten for clarity. (Lines: 39,40,41,42).

ii) Please revise the paragraph on engagement intervention (Lines 36, 37, 38). The abstract needs a summary of the results of the engagement intervention. The sentence provided only information on how the researchers assessed participants’ understanding of the ICF but not the results of the engagement intervention.

Done: The abstract has been reworded to improve the text's coherence. We also reworded the results paragraphs. The participants demonstrated their comprehension through cognitive processes, which were identified through our observations of the interactions during the educational intervention. We now also include a reference to the concept of comprehension adopted in the study, based on the Philosophy of Difference as articulated by Deleuze [1]. (Lines: 42,43,44,45,46,47,48).

2-INTRODUCTION 2.1-The abbreviation CHOMs has to be written in full in its first mention. (Line 64)

Done: The abbreviation CHOMs was incorrect. It has been corrected to CHIS (Controlled Human Infection Studies). (Lines:77).

2.2-The Educational Intervention. The first sentence (paragraph) has to be revised for clarity and comprehension. (Lines: 132, 133, 134)

Done: The section has been rewritten for clarity and understanding. (2.3- The Education Intervention -Lines:208,209,210,2011,2012,2013,2014.)

2.3-“Orality” and “Problematization” have to be defined in context to this research for comprehension by the reader. The reader has to understand what the researchers meant by “Orality” and “Problematization”. Please revise. (Lines: 138, 139)

Done: The concepts of orality and problematization have been described in the manuscript in separate paragraphs. (Lines:209,219,220,221).

3-RESULTS For the various issues addressed in this section, a numerical presentation of the results has to be given. 

E.g. Considering “Acceptability or Desire to participate”, the results have to present to the reader how many participants of the 95 expressed distinct positions? And this goes for all other sections under results.

 The definitions for G1, G2, G3, G4 – G9 have been added: G1, G2, G3, G4- G9 refers to the abbreviations of group 1, group 2- group 9. We had nine groups that participated in the education intervention.

No statistical analysis was carried out. We opted for content analysis with a qualitative approach without using statistical analysis. Even though it is highly productive, the studies in which Bardin's content analysis is used from a quantitative perspective are supported by the positivist paradigm, which values quantification and objectivity over the qualitative exploration of messages. Our study is qualitative; therefore, validity and scientific rigor differ from the quantitative approach but are based on equally rigorous principles and rules. As the referee makes clear, there are two types of analysis: Bardin (1977, p. 114) explains that quantitative analysis is based on the frequency of appearance of some aspects of the message, while qualitative analysis "[...] uses non-frequency indicators that can allow inferences to be made; for example, presence (or absence) can be as (or more) fruitful an index than frequency of appearance". Also, according to the referee, regardless of the type of analysis chosen, the required procedures include the preparation and exploration of the material, involving the operations of coding and breaking down the material collected into recording units. To comply with the request to make the analysis transparent, we have drawn up a table that summarizes the recording units, context units, and categories extracted from the study's empirical material stored in Excel spreadsheets. Due to the regional vocabulary used by the participants, it is challenging to translate into English, as this could alter the meaning of the original speeches.

Qualitative analysis was chosen for two reasons. First, the study's objective is centered around education and contemporary cognitive sciences. These are phenomena that, to be interpreted, require the monitoring and recording of pedagogical journeys rather than the outcome. Pedagogical paths, in turn, necessarily include interactions between those involved in the process. This methodological approach has broadened its horizons and acquired new dimensions in contemporary education studies. Recent research, events, and publications in this area have focused on new ways of conceiving and understanding the analysis of educational and cognitive processes in contemporary times[2]. The second reason for choosing the method lies in the data collection strategies. Education interventions and focus groups are tools that have in common the fact that they favor group interactions. In both the intervention and the focus group, some dialogues and discussions favor the production of knowledge by the participants. During group discussions, a collective census is always established, meanings are negotiated, and knowledge is produced because of these processes of social interaction between people. The data that emerges, therefore, is a product of group interaction. To ensure that it represents the group's thinking, we systematically ask the participants to agree with or refute the arguments presented, which confers reliability on the data obtained.

Response to Reviewer # 2

The manuscript was well written and scientifically sound, however, the authors had these few issues to address:

1. Provided result on the demographic and clinical characteristic of the study participants in the manuscript

Done: As explained in the manuscript, we do not have the participants' clinical data, as the study is limited to analyzing the participants' perceptions of the EIHC study. It was unnecessary to assess the participants' clinical conditions to collect their perceptions. The demographic data is described in lines 251,252,253,254 and 255 in the results section. Gender, age, and occupation were collected for this project. We also collected data on whether or not they had previously participated in clinical studies.

2. Under methodology, describe the method used to identify the hookworm infection

Answer: The project gathered the residents' perceptions of the endemic region through an educational intervention activity. It is well known that the study area has active hookworm transmission, but it was not an objective of our study to identify infected individuals.

3. Provide information on the ethical approval for the study

Done: Ethical approval information was inserted into the manuscript. A subtitle was created for ethical issues, 2.4 (Ethical aspects). This section describes all the ethical problems related to the study. 

4. Provide information on the inclusion and exclusion criteria used for the study. 

Done: The inclusion and exclusion criteria were inserted into the manuscript in the methodology part. (Lines: 127,128,128,130,131,132).

5. Provide a flow chart for the study showing the number participant recruited to the end of the study. Provide information on those who exited from the follow and the reasons.

Not done: The study involved one specific educational intervention activity, which took place with nine groups. One hundred forty-one people were invited, and 95 people attended the activity. These 95 people participated in the activity, which lasted approximately 1 hour. Since there was only one group meeting for each participant, all individuals included completed the activity and stayed in the study; hence, there was no loss to follow up. 

Response to reviewer # 3

1-The statistical analysis performed was not appropriate and rigorous because even though the researchers used Bardin's Content Analysis for their data analysis but failed to categorize or codify the words, themes, and concepts within the text that would enable them to subject their data to quantitative analysis. Content analysis can be both quantitative (focused on counting and measuring) and qualitative (focused on interpreting and understanding). In both types, you categorize or “code” words, themes, and concepts within the texts and then analyze the results. It is therefore recommended that the authors do the needful and subject their data to quantitative analysis.

Done: No statistical analysis was carried out. We opted for content analysis with a qualitative approach without using statistical analysis. Even though it is highly productive, the studies in which Bardin's content analysis is used from a quantitative perspective are supported by the positivist paradigm, which values quantification and objectivity over the qualitative exploration of messages. Our study is qualitative; therefore, validity and scientific rigor differ from the quantitative approach but are based on equally rigorous principles and rules. As the referee makes clear, there are two types of analysis: Bardin (1977, p. 114) explains that quantitative analysis is based on the frequency of appearance of some aspects of the message, while qualitative analysis "[...] uses non-frequency indicators that can allow inferences to be made; for example, presence (or absence) can be as (or more) fruitful an index than frequency of appearance". Also, according to the referee, regardless of the type of analysis chosen, the required procedures include the preparation and exploration of the material, involving the operations of coding and breaking down the material collected into recording units. To comply with the request to make the analysis transparent, we have drawn up a table that summarizes the recording units, context units, and categories extracted from the study's empirical material stored in Excel spreadsheets. Due to the regional vocabulary used by the participants, it is difficult to translate into English, as this could alter the meaning of the original speeches.

Qualitative analysis was chosen for two reasons. First, because of the nature of the object of study. It is located in the field of education and contemporary cognitive sciences. These are phenomena that, to be interpreted, require the monitoring and recording of pedagogical journeys rather than the outcome. Pedagogical paths, in turn, necessarily include interactions between those involved in the process. This methodological approach has broadened its horizons and gained new dimensions in contemporary education studies. Recent research, events, and publications in this area have focused on new ways of conceiving and understanding educational and cognitive process analysis in contemporary times[2]. The second reason for choosing the method lies in the data collection strategies. Education interventions and focus groups are tools that have in common the fact that they favor group interactions. In both the intervention and the focus group, some dialogues and discussions favor the production of knowledge by the participants. During group discussions, a collective census is always established, meanings are negotiated, and knowledge is produced due to these processes of social interaction between people. The data that emerges, therefore, is a product of group interaction. To e

---

## [Decision Letter · Decision Letter 1]

5 Feb 2024

Public Engagement for the Conduct of a Controlled Human Infection Study testing vaccines against Necator americanus (hookworm) in areas of active hookworm transmission in Brazil.

PONE-D-23-25173R1

Dear Dr. Ribeiro,

We’re pleased to inform you that your manuscript has been judged scientifically suitable for publication and will be formally accepted for publication once it meets all outstanding technical requirements.

Kind regards,

Kenneth Bentum Otabil, PhD

Academic Editor

PLOS ONE

Additional Editor Comments (optional):

Reviewers' comments:

Reviewer's Responses to Questions

**Comments to the Author**

1. If the authors have adequately addressed your comments raised in a previous round of review and you feel that this manuscript is now acceptable for publication, you may indicate that here to bypass the “Comments to the Author” section, enter your conflict of interest statement in the “Confidential to Editor” section, and submit your "Accept" recommendation.

Reviewer #2: All comments have been addressed

2. Is the manuscript technically sound, and do the data support the conclusions?

Reviewer #2: Yes

3. Has the statistical analysis been performed appropriately and rigorously? 

Reviewer #2: N/A

4. Have the authors made all data underlying the findings in their manuscript fully available?

Reviewer #2: Yes

5. Is the manuscript presented in an intelligible fashion and written in standard English?

Reviewer #2: Yes

6. Review Comments to the Author

Reviewer #2: No comments, all issue raised has been answered

The manuscript is now in good shape for publication.

7. PLOS authors have the option to publish the peer review history of their article (what does this mean?). If published, this will include your full peer review and any attached files.

Reviewer #2: **Yes: **Dr Benjamin Amoani

---

## [Editor Report · Acceptance letter]

9 May 2024

PONE-D-23-25173R1 

PLOS ONE

Dear Dr. Ribeiro, 

I'm pleased to inform you that your manuscript has been deemed suitable for publication in PLOS ONE. Congratulations! Your manuscript is now being handed over to our production team.

Kind regards, 

on behalf of

Dr. Kenneth Bentum Otabil 

Academic Editor

PLOS ONE